# A structural mechanism for bacterial autotransporter glycosylation by a dodecameric heptosyltransferase family

Qing Yao[1†], Qiuhe Lu[1†], Xiaobo Wan[2], Feng Song[3,4], Yue Xu[1], Mo Hu[5,6], Alla Zamyatina[7], Xiaoyun Liu[5,6], Niu Huang[2], Ping Zhu[3]*, Feng Shao[1,3]*

[1]Dr Feng Shao's Laboratory, National Institute of Biological Sciences, Beijing, China; [2]Dr Niu Huang's Laboratory, National Institute of Biological Sciences, Beijing, China; [3]National Laboratory of Biomacromolecules, Institute of Biophysics, Chinese Academy of Sciences, Beijing, China; [4]Shandong Provincial Key Laboratory of Functional Macromolecular Biophysics, Institute of Biophysics, Dezhou University, Dezhou, China; [5]Institute of Analytic Chemistry, College of Chemistry and Molecular Engineering, Peking University, Beijing, China; [6]Synthetic Functional Biomolecules Center, College of Chemistry and Molecular Engineering, Peking University, Beijing, China; [7]Department of Chemistry, University of Natural Resources and Life Sciences, Vienna, Austria

*For correspondence: zhup@ ibp.ac.cn (PZ); shaofeng@nibs. ac.cn (FS)

†These authors contributed equally to this work

## Abstract

A large group of bacterial virulence autotransporters including AIDA-I from diffusely adhering *E. coli* (DAEC) and TibA from enterotoxigenic *E. coli* (ETEC) require hyperglycosylation for functioning. Here we demonstrate that TibC from ETEC harbors a heptosyltransferase activity on TibA and AIDA-I, defining a large family of bacterial autotransporter heptosyltransferases (BAHTs). The crystal structure of TibC reveals a characteristic ring-shape dodecamer. The protomer features an N-terminal β-barrel, a catalytic domain, a β-hairpin thumb, and a unique iron-finger motif. The iron-finger motif contributes to back-to-back dimerization; six dimers form the ring through β-hairpin thumb-mediated hand-in-hand contact. The structure of ADP-D-glycero-β-D-manno-heptose (ADP-D,D-heptose)-bound TibC reveals a sugar transfer mechanism and also the ligand stereoselectivity determinant. Electron-cryomicroscopy analyses uncover a TibC–TibA dodecamer/hexamer assembly with two enzyme molecules binding to one TibA substrate. The complex structure also highlights a high efficient hyperglycosylation of six autotransporter substrates simultaneously by the dodecamer enzyme complex.

## Introduction

Protein glycosylation is one of the most abundant post-translational modifications in all domains of life (*Spiro, 2002*). Recent studies have appreciated protein glycosylation in bacteria, which is often associated with pathogen virulence and immune modulation (*Szymanski and Wren, 2005*; *Abu-Qarn et al., 2008*; *Nothaft and Szymanski, 2010*). For instance, flagella glycosylation is found in many bacteria species including *Campylobacter jejuni*, *Helicobacter pylori*, *Clostridium spp.* and *Pseudomonas aeruginosa*, contributing to bacterial locomotion or virulence (*Schirm et al., 2003*; *Nothaft and Szymanski, 2010*). Recently, we and others have shown that secreted effectors from enteropathogenic *Escherichia coli* (EPEC) and related enteric pathogens harbor an arginine *N*-acetylglucosamine transferase activity that modifies host death-domain proteins and is essential for bacterial colonization in infected mice (*Li et al., 2013*; *Pearson et al., 2013*).

The autotransporter secretion pathway (also known as the type V secretion system) in bacteria delivers autotransporters onto the bacterial surface. Autotransporters, representing the largest

**eLife digest** Bacteria release proteins known as virulence factors to help them infect host cells. Many bacteria are surrounded by two membranes, so virulence factors must be able to pass through both of these membranes.

Autotransporters are a group of virulence factors that pass through the inner membrane and anchor themselves in the outer membrane; this allows part of the autotransporter to project from the surface of the bacterial cell and stick to the surface of the cell that the bacterium is about to infect. Many autotransporters are coated with sugar molecules and this increases their ability to adhere to cells. Enzymes called glycosyltransferases ensure that this sugar coating process takes place.

Autotransporters contain two sections: the passenger domain and the beta domain. The passenger domain is important for virulence, while the beta domain forms a pore in the outer membrane that the passenger domain passes through to reach the outer surface of the bacterium. TibA and AIDA-1 are autotransporters associated with two types of bacteria that infect the intestines and cause diarrhea. Similar autotransporters are found in a wide range of bacteria, but the precise details of how these autotransporters are coated with sugar molecules are not fully understood.

Yao et al. now show that a glycosyltransferase called TibC, which is found in many bacteria, adds large numbers of sugar molecules to the passenger domains of both TibA and AIDA-1. To learn more about this process Yao et al. used X-ray diffraction to work out the structure of TibC. Strikingly, this revealed that TibC proteins come together to form a large circular structure that contains two rings, each made of six TibC proteins. The integrity of this structure is maintained by the presence of iron atoms, which also gives TibC a characteristic brown colour.

Yao et al. also studied what happens when TibC binds to TibA; a technique called electron cryo-microscopy revealed that six TibA molecules are distributed along the inner surface of the circular TibC structure, with each TibA protein binding to two TibC proteins. This arrangement allows for the efficient transfer of sugar molecules from the glycosyltransferase to the autotransporter.

family of bacterial virulence factors, share a similar structural organization containing a signal peptide followed by a functional passenger domain and a C-terminal β-barrel translocation domain. Autotransporters play a critical role in diverse aspects of bacterial physiology including proteolytic digestion of host proteins, biofilm formation, adhesion and invasion of host cells, and intracellular motility (*Henderson et al., 2004*; *Lazar Adler et al., 2011*; *Wells et al., 2007*). A subfamily of autotransporters, including AIDA-I (adhesin involved in diffuse adherence) from diffusely adhering *E. coli* (DAEC) 2787 (*Benz and Schmidt, 1989*) and TibA from enterotoxigenic *E. coli* (ETEC) H10407 (*Elsinghorst and Weitz, 1994*), are glycosylated in their passenger domains (*Lindenthal and Elsinghorst, 1999*; *Benz and Schmidt, 2001*; *Sherlock et al., 2006*), which functions in bacterial auto-aggregation and adhesion to host cells (*Benz and Schmidt, 1989*; *Sherlock et al., 2004*; *Charbonneau and Mourez, 2007*). AIDA-I and TibA-like autotransporters are present in diverse bacterial species. Previous studies carried out in the ectopic system (*Benz and Schmidt, 2001*; *Moormann et al., 2002*) indicate a role for AAH (autotransporter adhesin heptosyltransferase) in DAEC and its ETEC homologue TibC in AIDA-I and TibA glycosylation. However, AAH and TibC harbor no sequence homology to known glycosyltransferases and there has been no biochemical evidence demonstrating their glycosyltransferase activity.

In a separate parallel study (*Lu et al., 2014*) we showed that AAH is a *bona fide* heptosyltransferase belonging to a large bacterial autotransporter heptosyltransferase (BAHT) family. Here we determine the crystal structures of TibC heptosyltransferase, both alone and in complex with ADP-D-glycero-β-D-manno-heptose (ADP-D,D-heptose). The structure shows a symmetric ring-shape dodecamer. The protomer features a β-hairpin thumb and an iron-finger motif, both required for the dodecamer assembly. The ligand-bound structure reveals a sugar transfer mechanism and determinants for the ligand stereoselectivity for TibC and AAH. Furthermore, electron cryomicroscopy (cryo-EM) analyses of a TibC–TibA dodecamer/hexamer enzyme–substrate complex reveal the structural basis for high efficient autotransporter hyperheptosylation by the TibC dodecamer.

## Results

### TibC catalyzes AIDA-I/TibA heptosylation and confers AIDA-I-mediated bacterial adhesion to host cells

We first confirmed the autotransporter heptosyltransferase activity of TibC and its function in mediating bacterial adhesion to host cells (*Lu et al., 2014*). Co-expression of TibC or AAH together with AIDA-I in *E. coli* BL21 cells resulted in a tight adhesion of the bacteria to HeLa cells (*Figure 1A*) and AIDA-I glycosylation (*Figure 1B*). Co-expression of TibC also induced TibA glycosylation in the heterologous system (*Figure 1C*). We further identified a minimal passenger domain fragment of TibA (TibA$_{305-350}$) that could be efficiently glycosylated by TibC (*Figure 1D,E*). TibA$_{305-350}$ tagged with Flag epitopes at both termini was then fused C-terminal to the Small Ubiquitin-like Modifier (SUMO). The SUMO-Flag-TibA$_{305-350}$-Flag fusion protein showed a slower migration on the SDS-PAGE gel when co-expressed with TibC (*Figure 1F*). Electrospray ionization (ESI) mass spectrometry analysis gave a mass of 20,758 Da that matched the theoretical mass (20,757 Da) of SUMO-Flag-TibA$_{305-350}$-Flag (*Figure 1G*). Co-expression of TibC resulted in mass increase of 576, 768, or 960 Da, corresponding to the addition of 3, 4, and 5 heptoses, respectively (*Figure 1G*). The modified fusion protein was further digested with the endoproteinase Asp-N for liquid chromatography-tandem mass spectrometry (LC-MS/MS) analysis. Asp-N digestion is expected to yield two major peptide fragments, DK-$_{305}$SASKVIQNSGGAVITNTSAAVSGTN$_{329}$ and $_{330}$DNGSFSIAGGSAVNMLLENGG$_{350}$. We were able to detect abundant signals for both unmodified and modified DK-$_{305}$SASKVIQNSGGAVITNTSAAVSGTN$_{329}$ in full scan MS analyses and only weak signals of the modified $_{330}$DNGSFSIAGGSAVNMLLENGG$_{350}$ peptide, probably due to the hydrophobic nature. Mass measurements indicated that both peptides had multiple heptose modifications; a mixed population of modified peptides, containing different numbers of heptose conjugations, was observed. Similarly to that performed with AAH modification of AIDA-I (*Lu et al., 2014*), serine residues (Ser-313 and Ser-322 within DK-$_{305}$SASKVIQNSGGAVITNTSAAVSGTN$_{329}$), but not threonine residues, were identified to be the modification sites by electron transfer dissociation (ETD) mass spectrometry (*Figure 1H*). Supporting the heptosyltransferase activity of TibC, recombinant TibC was found to directly modify synthetic peptides derived from the passenger domains of either AIDA-I or TibA in which ADP-D,D-heptose, but not the anomer ADP-L,D-heptose, could serve as the sugar ligand (*Figure 1I*).

AAH and TibC homologues, defined as the BAHT family (*Lu et al., 2014*), are widely present in pathogenic bacteria species including *Citrobacter rodentium*, *Salmonella enterica* serovar Urbana, *Shigella sp.* D9, *Laribacter hongkongensis*, *Cronobacter sakazakii*, and several *Burkholderia* species (*Figure 1—figure supplement 1*). These proteins generally bear more than 50% sequence homology to TibC/AAH and contain residues critical for the heptosyltransferase activity (see below). In the genomic locus, genes encoding BAHT are often followed by an autotransporter or a putative autotransporter, consistent with the notion that BAHT modifies its autotransporter partner for functioning (*Lu et al., 2014*). Consistent with the modification of AIDA-I by TibC, the passenger domain of AIDA-I shows sequence similarities to that of TibA and contains a number of heptosylation motifs identified in TibA previously (*Lu et al., 2014*) (*Figure 1—figure supplement 2*).

### Crystal and cryo-EM structures of TibC dodecamer

To reveal the mechanism of the BAHT family, we attempted to solve the crystal structure. Among several BAHTs analyzed, TibC behaved the best in recombinant expression and homogeneity. Purified His$_6$-TibC was crystallized, but the good-looking crystals did not diffract due to internal disorders. TibC obtained from glutathione-S-transferase (GST)-fusion expression was also crystallized, which only diffracted to 8 Å despite extensive optimizations. A number of point mutations designed to reduce the surface entropy were screened and a TibC variant with three mutated epitopes (E83A/E84A, K400A/K401A, and Q215A/E216A) produced the highest quality crystal. This mutant exhibited identical biochemical and biological functions to wild-type TibC (*Figure 1A–C*). A 2.9 Å structure was solved by single wavelength dispersion phasing using anomalous signals from selenomethionine (SeMet) and the naturally bound ferric ion (*Table 1*).

Strikingly, one asymmetric unit in the solved structure contains 12 TibC molecules, which are assembled into a large ring structure (*Figure 2A*). The overall shape of the dodecamer ring resembles a circular garland with an external diameter of 145 Å and a height of 72 Å. Parts of the 12 protomers are lined side by side at the central plate to form the stem of the garland with an inner diameter of

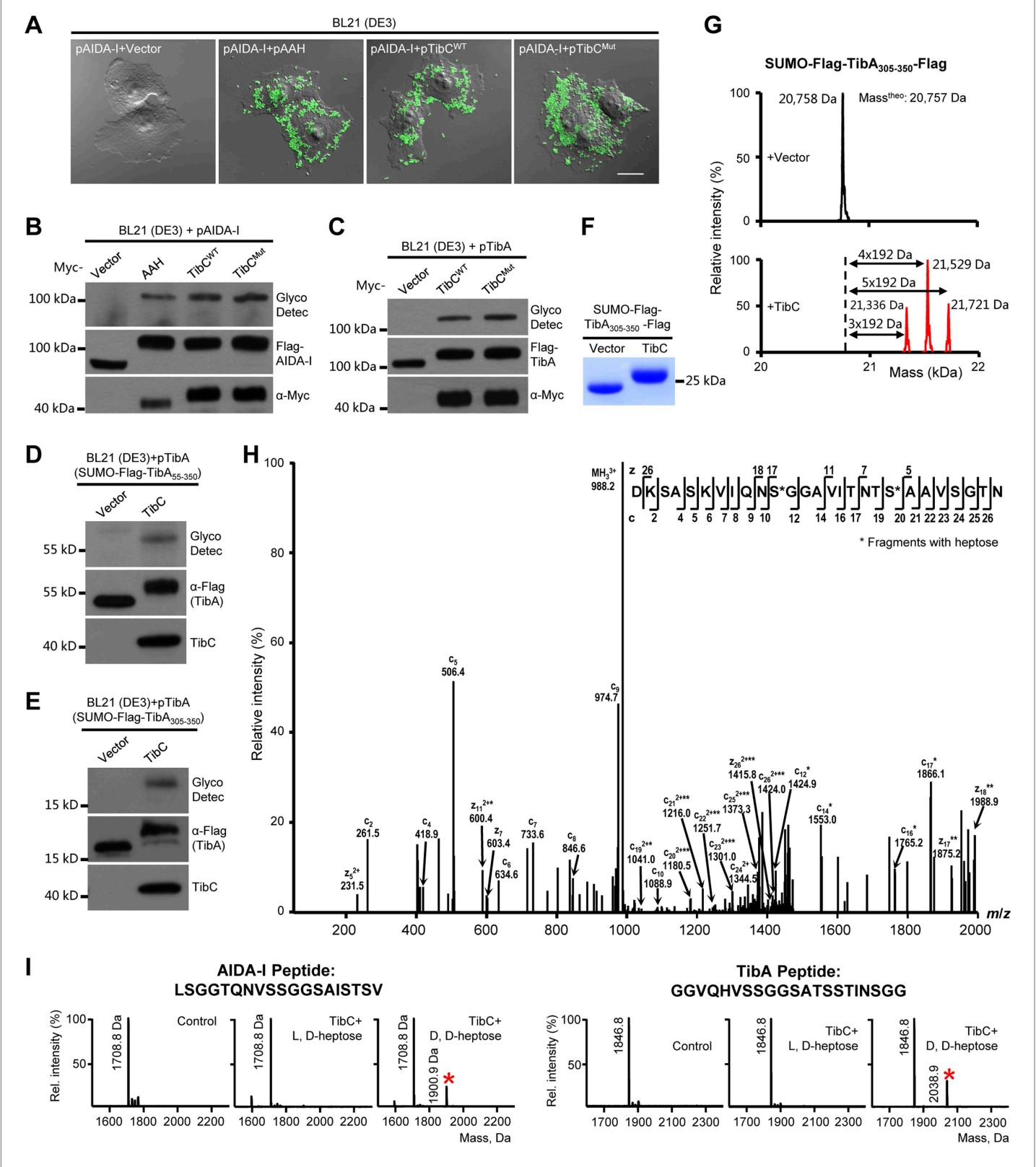

**Figure 1**. TibC catalyzes TibA/AIDA-I heptosylation and confers AIDA-I-mediated bacterial adhesion to host cells. (**A**) *E. coli* BL21 expressing EGFP was transformed with a plasmid harboring Flag-AIDA-I (pAIDA-I) together with a vector or Myc-tagged autotransporter adhesin heptosyltransferase (AAH)/TibC. TibC^WT, wild-type TibC; TibC^mut, the TibC mutant used for crystallization. Infected cells and the bacteria were visualized by differential

*Figure 1. Continued on next page*

*Figure 1. Continued*

interference contrast microscopy and the green fluorescence, respectively. Scale bar 20 µm. (**B** and **C**) Assays of TibC-catalyzed AIDA-I and TibA glycosylation. Lysates of *E. coli* BL21 (DE3) cells transformed with indicated constructs were subjected to anti-Flag and -Myc immunoblotting or glycosylation analysis. The Flag tag in AIDA-I/TibA was inserted between the signal peptide and the passenger domain. (**D–F**) Indicated TibA truncations fused C-terminal to the SUMO-Flag were co-expressed with a vector or Myc-TibC in *E. coli* BL21. Shown in (**F**) is Coomassie brilliant blue staining of purified SUMO-Flag-TibA$_{305-350}$-Flag. (**G**) ESI mass spectrometry analyses of TibC modification of SUMO-Flag-TibA$_{305-350}$-Flag in the *E. coli* system. The red peaks mark the glycosylated species. (**H**) ETD tandem mass spectrum of a triply charged Asp-N digested peptide from TibC-modified SUMO-Flag-TibA$_{305-350}$-Flag. The fragmentation patterns generating the observed c and z ions are illustrated along the peptide sequence on top of the spectrum. Asterisk marks a modification on the serine by a heptose. (**I**) In vitro heptosyltransferase activity of TibC. Two peptides derived from the passenger domains of AIDA-I (left) and TibA (right), respectively, were left untreated (control) or reacted with TibC in the presence of indicated sugar ligands. Shown are MALDI-TOF mass spectra of the reacted peptides (*, peptides modified by one heptose).
The following figure supplements are available for figure 1:

**Figure supplement 1**. Multiple sequence alignment of the BAHT Family.

**Figure supplement 2**. Amino acid sequence of AIDA-I passenger domain (residues 58–588) arranged by β-helix repeat units.

~110 Å (*Figure 2A*). The inner surface of the stem is decorated with uniformly distributed 12 ferric ions, forming a characteristic iron belt along the ring (*Figure 2A*). The presence of ferric ions is expected from the brownish color of recombinant TibC protein in solution. Six symmetric TibC protomers project upwards from the central plate while the other six project downwards, generating a gear-like shape at both sides. Two adjacent TibC protomers projecting oppositely contact each other in a back-to-back manner around a dyad axis; six back-to-back dimers are further linked together in a hand-in-hand fashion around a sixfold axis to form the garland-like ring (*Figure 2B*). Thus, the dodecamer assembly can be viewed as a hexameric oligomerization of dimers. The back-to-back interface is extensive and tilts across the central plate by 45° (*Figure 2A,B*). The hand-in-hand connection creates 12 tilted grooves extending to the interior of the garland from both up and down sides.

To investigate whether the dodecamer observed in the crystal structure results from crystal packing, single particle cryo-EM analyses of highly purified TibC protein were performed (*Figure 2—figure supplement 1*). A 3D reconstruction of 11.5 Å was obtained from the cryo-EM images, which showed a remarkably similar dodecamer architecture to that determined by X-ray crystallography (*Figure 2C*). This suggests that the dodecamer ring architecture is the predominant form of TibC protein in solution and likely represents a functional physiological state. Consistently, purified TibC eluted as an oligomer of ~500 kDa on the gel filtration column (see below) and the molecular weight was determined to be 578 kDa by analytic ultracentrifugation (*Figure 2—figure supplement 2*).

## Structure of TibC protomer and the glycosyltransferase domain

All the 12 TibC protomers adopt an identical structure containing an N-terminal β-barrel domain, a core catalytic domain, a β-hairpin thumb, and an iron-finger motif which are arranged into a palm-like shape (*Figure 3A*). The β-hairpin thumb and the iron-finger motif are insertions into the catalytic domain. The flattened β-barrel (residues 1–97) is structured from two four-stranded sheets that pack against each other. The barrel is stacked against the catalytic domain from the side and makes extensive contacts with β9, β12, and the loop linking α3 and β12; several hydrophobic residues from both domains are in close proximity at the interface (*Figure 3—figure supplement 1A*). The β-barrel is well separated from the catalytic domain and is unlikely to play a direct role in sugar transfer. Notably, deletion of the N-terminal 80 or 95 residues abolished TibC heptosylation of TibA (*Figure 3—figure supplement 1B*), indicating a possible function of the β-barrel domain in either substrate recognition or maintaining structural integrity of the enzyme.

The catalytic domain contains two lobes (residues 98–195 and 218–406, respectively) linked by a long loop (residues 196–217). The N-lobe, containing β9–12, α1–4 and surrounding loops, adopts a compact Rossmann-like globular fold. Different from the classical Rossmann fold that has α-helices flanking the central β-sheet, the β-sheet in the N-lobe is flanked by α-helices (α1–4) at one side but the β-barrel at the other side (*Figure 3A*). The C-lobe (α5–11 and β13–18) features a central β-sheet (β13 and β16–18) sandwiched by α7/α8 at one side and α6/α11 at the other side, adopting a typical Rossmann fold.

**Table 1.** Data collection and refinement statistics

| Crystals | TibC-SeMet | TibC D110A in complex with D,D-heptose |
|---|---|---|
| Data collection | | |
| Space group | $P2_1$ | $P2_1$ |
| Wavelength (Å) | 0.9789 | 0.9792 |
| a, b, c (Å) | 87.8, 314.4, 164.5 | 83.3, 313.2, 164.7 |
| α, β, γ (°) | 90, 101.4, 90 | 90, 101.3, 90 |
| Resolution range (Å)* | 20.0–2.90 (2.95–2.90) | 20.0–3.87 (3.94–3.87) |
| No. of unique reflections | 194,303 (9760) | 79,888 (3968) |
| Completeness (%) | 99.9 (99.9) | 99.9 (100) |
| Redundancy | 7.3 (6.1) | 4.6 (4.6) |
| I/σI | 19.5 (2.6) | 9.18 (2.0) |
| $R_{merge}$ (%) | 12.7 (98.6) | 23.7 (92.8) |
| Refinement statistics | | |
| $R_{work}$/$R_{free}$ (%)† | 20.8/24.4 | 26.4/27.6 |
| No. of protein atoms | 37,201 | 37,136 |
| No. of ligands atoms | 32 | 504 |
| No. of waters | 5 | 0 |
| RMSD bond lengths (Å) | 0.013 | 0.006 |
| RMSD bond angles (°) | 1.40 | 1.11 |
| Average overall B-factor | 69.10 | 150 |
| Iron atoms B-factor | 58.90 | 150 |
| ADP-heptose B-factor | | 150 |
| Ramachandran plot statistics | | |
| Most favored regions (%) | 94.51 | 94.73 |
| Additional allowed regions (%) | 5.36 | 4.45 |
| Outlier regions (%) | 0.13 | 0.83 |

*The data for the highest resolution shell are shown in parentheses.

†$R_{free}$ is calculated by omitting 5% of the total number of reflections in model refinement.

RMSD, root-mean-square deviation; SeMet, selenomethionine;

The catalytic cores of nearly all known glycosyltransferases, despite the extraordinary sequence divergence, converge onto two general folds—namely, GT-A and GT-B (*Unligil and Rini, 2000*; *Breton et al., 2006*; *Lairson et al., 2008*; *Breton et al., 2012*). The typical GT-A fold consists of two α/β/α domains with a continuous central β-sheet. The GT-A glycosyltransferase features a D×D motif that binds to the nucleotide phosphate in the sugar donor via a metal ion (*Bourne and Henrissat, 2001*). The GT-B fold has two Rossmann-like domains separated by a cleft, in which a sugar donor is positioned for nucleophilic attack by the substrate. The GT-B glycosyltransferase does not require metal ion for catalyzing sugar transfer (*Wrabl and Grishin, 2001*; *Hu and Walker, 2002*). The catalytic domain of TibC structurally more resembles the GT-B fold such as WaaC and MurG, two bacterial glycosyltransferases involved in cell wall synthesis (*Hu et al., 2003*; *Grizot et al., 2006*), with its own distinctions (*Figure 3B*). One major difference is that the N-lobe of the TibC catalytic domain is not a typical Rossmann fold; its central β-sheet is four-stranded rather than six-stranded for a Rossmann fold and also misses the flanking α-helices at one side (*Figure 3B*).

### The β-hairpin thumb and hand-in-hand contact

The TibC catalytic core is accessorized by two unique structural modules, the β-hairpin thumb and the iron-finger motif, both of which are important for the dodecamer assembly. The β-hairpin thumb consists of β15, β16, and a four-residue hairpin loop projecting away from the globular C-lobe (*Figure 3A*). The β-hairpin thumb mediates the hand-in-hand contact between adjacent dimers (*Figure 2B*).

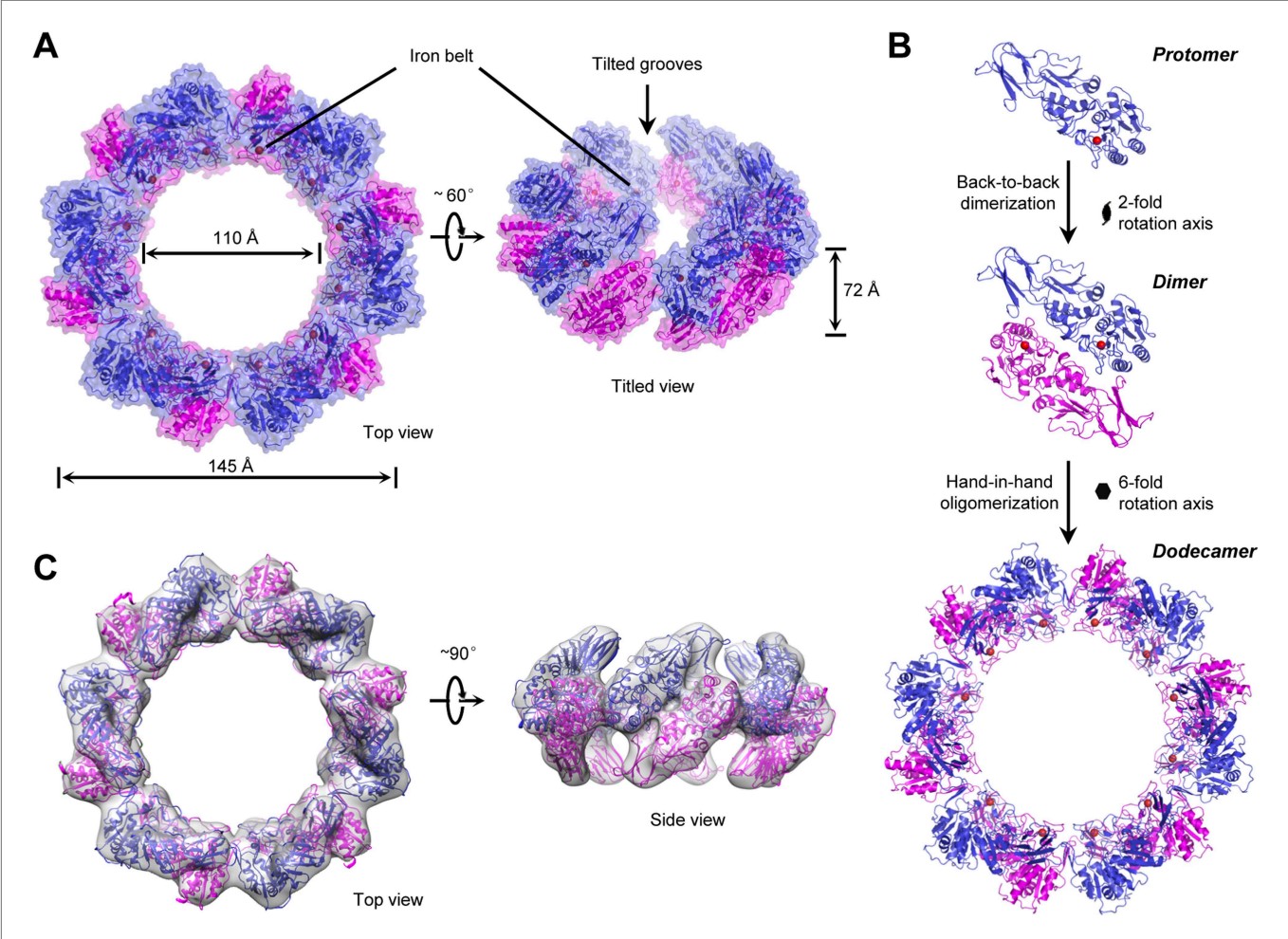

**Figure 2**. The overall structure and assembly of the TibC dodecamer. (**A**) The surface view of the TibC dodecamer. The two adjacent symmetric protomers are colored blue and magenta, respectively. The ferric ions are shown as red spheres. Shown on the left is a top view along the sixfold axis and on the right is a tilted view. (**B**) Assembly of the TibC dodecamer. Two adjacent protomers form a back-to-back dimer with a twofold symmetry. Six symmetric dimers are further assembled through hand-in-hand contacts into a large ring. (**C**) Cryo-EM reconstruction of TibC at 11.5 Å resolution. The crystal structure was fit into the cryo-EM envelope. Shown are top and side views of the EM structure.

The following figure supplements are available for figure 2:

**Figure supplement 1**. Cryo-EM images and 3D reconstruction of the TibC dodecamer.

**Figure supplement 2**. Analytical ultracentrifugation sedimentation velocity analysis of TibC dodecamer and TibC–TibA dodecamer/hexamer complex.

Phe-265 and Val-266 at the tip of the thumb, together with a nearby Phe-368 from the catalytic domain, contact the same set of residues from the adjacent protomer to form a hydrophobic cluster (*Figure 4A*). To validate the structural observations, the three residues were each subjected to mutational analyses. TibC F368D mutant was largely insoluble. Both F265D and V266D mutants eluted as a dimer from the gel filtration column (*Figure 4B*) and failed to catalyze TibA heptosylation in *E. coli* (*Figure 4C*). These results highlight the importance of the β-hairpin thumb in mediating the hand-in-hand association for assembly of an active TibC dodecamer. The data also explain the unique presence of two successive hydrophobic residues on TibC β-hairpin loop, which differs from other β-hairpin loops that are usually dominated by hydrophilic residues (*Sibanda and Thornton, 1985*). The β-hairpin thumb is highly conserved in the BAHT family (*Figure 1—figure supplement 1*), supporting the view that an intact dodecamer is critical for this family of heptosyltransferase to modify its partner autotransporter.

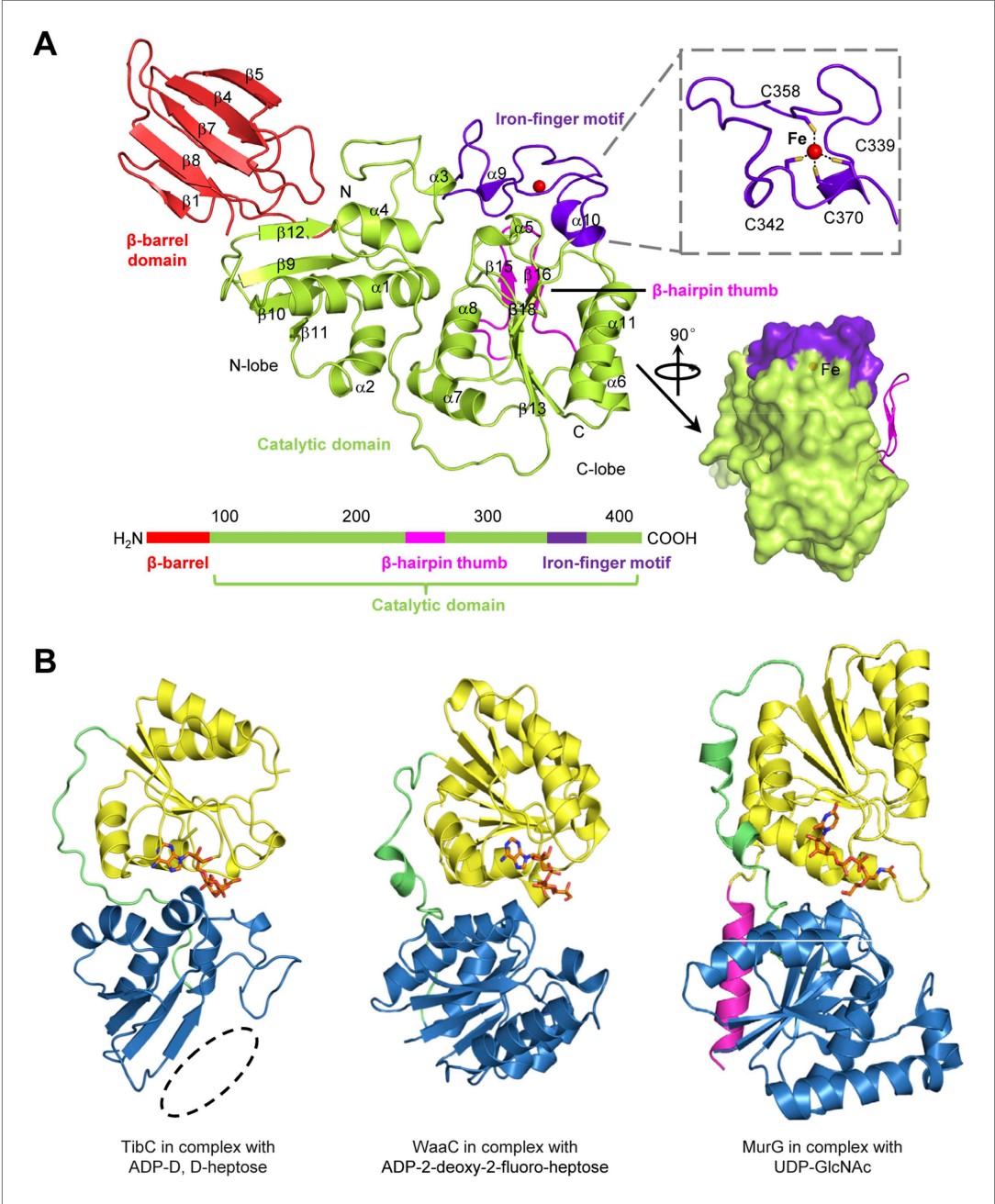

**Figure 3.** The TibC protomer structure and the glycotransferase domain. (**A**) Ribbon diagram of the TibC protomer structure. The upper right shows tetrahedral coordination of the ferric ion by four cysteine residues. Shown on the lower right is a lateral view with the β-hairpin thumb highlighted as magenta ribbons. The domain organization along the primary structure is on the lower left. (**B**) Structural comparison of TibC with GT-B glycosyltransferases. Left, the catalytic domain of ADP-D,D-heptose-bound TibC structure determined in this study; Center, WaaC bound with ADP-2-deoxy-2-fluoro-heptose (PDB ID: 2H1H); Right, MurG bound with UDP-GlcNAc (PDB ID: 1NLM). The overall structures and sugar ligands are in ribbon diagram and orange sticks, respectively. The N- and C-lobe are colored blue and yellow, respectively. The β-hairpin thumb insertion and the iron-finger motif in TibC are omitted for clarity. The linker between the two lobes and an extra C-terminal helix in MurG are in green and magenta, respectively. The dashed circle highlights the absence of α-helices at the N terminus of the N-lobe in TibC.

The following figure supplement is available for figure 3:

**Figure supplement 1.** Interaction between the β-barrel and the N-lobe of the catalytic domain in TibC protomer.

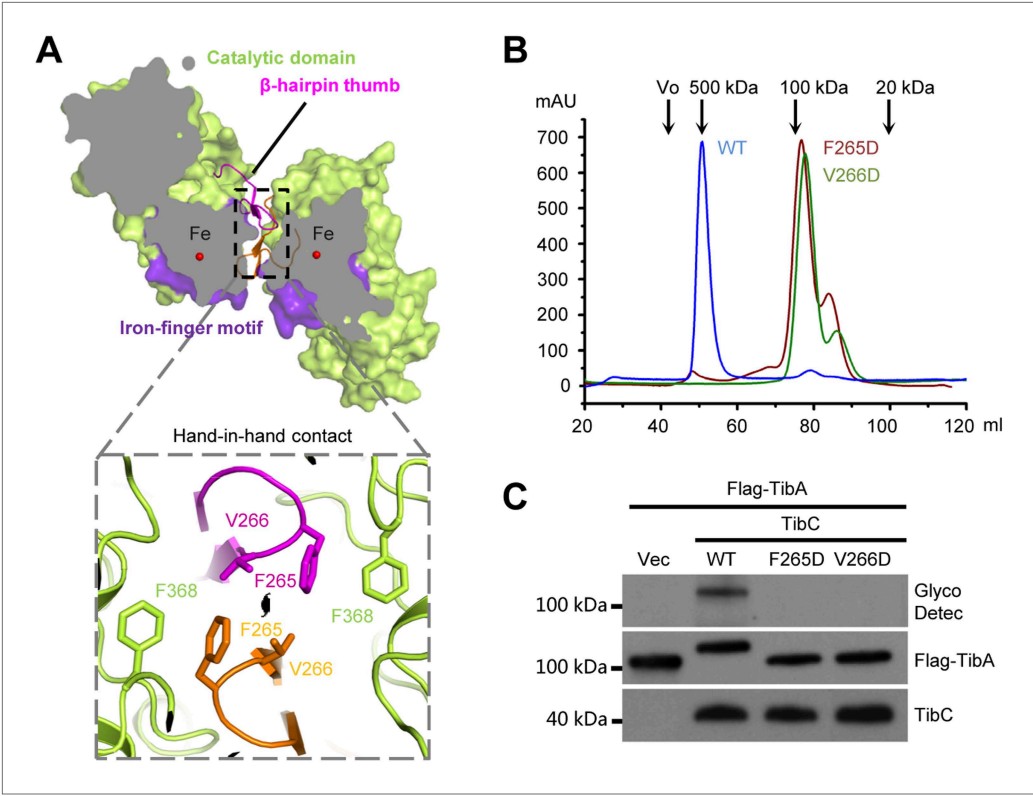

**Figure 4**. The β-hairpin thumb-mediated hand-in-hand contact required for TibC dodecamer assembly and catalytic activity. (**A**) β-hairpin thumb-mediated hand-in-hand contact. Upper, surface presentation of two adjacent TibC protomers with the β-hairpin thumb in ribbons. Lower, interface details with key residues in sticks. Black lens, the twofold axis. (**B** and **C**) Effects of hand-in-hand contact mutations on TibC dodecamer formation and catalyzing TibA glycosylation. TibC (WT or indicated mutant) proteins were loaded onto a gel filtration column in (**B**). Black arrows mark the molecular weight calibration. Vo, void volume. The experiments in (**C**) were performed and data are presented similarly to those in **Figure 1B**.

## A unique iron-finger motif and dodecamer assembly

The other insertion module (residues -368) features two knuckles with long exposed loops capping and embellishing the C-lobe (**Figure 3A**). Within the module, a ferric ion is tetrahedrally coordinated by four conserved cysteine residues (Cys-339, 342, 358, and 370 in TibC) (**Figure 3A** and **Figure 1— figure supplement 1**), analogous to the zinc-finger motif. This iron-finger motif lies ~20 Å from the catalytic cleft (see below) and its architecture belongs to the 1Fe-0S cluster. Differing from the most common zinc-finger motif, the 1Fe-0S iron-finger motif is only observed once in rubredoxin protein found in sulfur-metabolizing bacteria and archaea (**Lovenberg and Sobel, 1965**; **Adman et al., 1975**). In rubredoxin, the 1Fe-0S cluster acts as an electron transfer carrier (**Sieker et al., 1994**). The iron-finger motif in TibC, however, does not seem to function in electron transfer according to our structural and functional analyses. Notably, mutation of any of the four iron-coordinating cysteine residues resulted in a colorless TibC protein (**Figure 5A**) due to the loss of iron binding. The four cysteine mutants eluted from the gel filtration column as heterogeneous interconverting oligomers with their sizes smaller than a dodecamer (**Figure 5B**). Thus, the iron finger is required for maintaining structural integrity of the TibC dodecamer. Consistently, the four iron binding-deficient mutants all failed to catalyze TibA heptosylation in the *E. coli* system (**Figure 5C**).

Structurally, the iron-finger motif contributes to the back-to-back dimerization. A portion of the iron-finger motif, together with several surface residues from the catalytic domain of one protomer, makes extensive contacts with the β-barrel as well as the catalytic domain of the other protomer in the back-to-back dimer (**Figure 5D**). A symmetric interface is formed via the same kind of interactions between the two protomers. Extensive polar and hydrophobic contacts occur at the

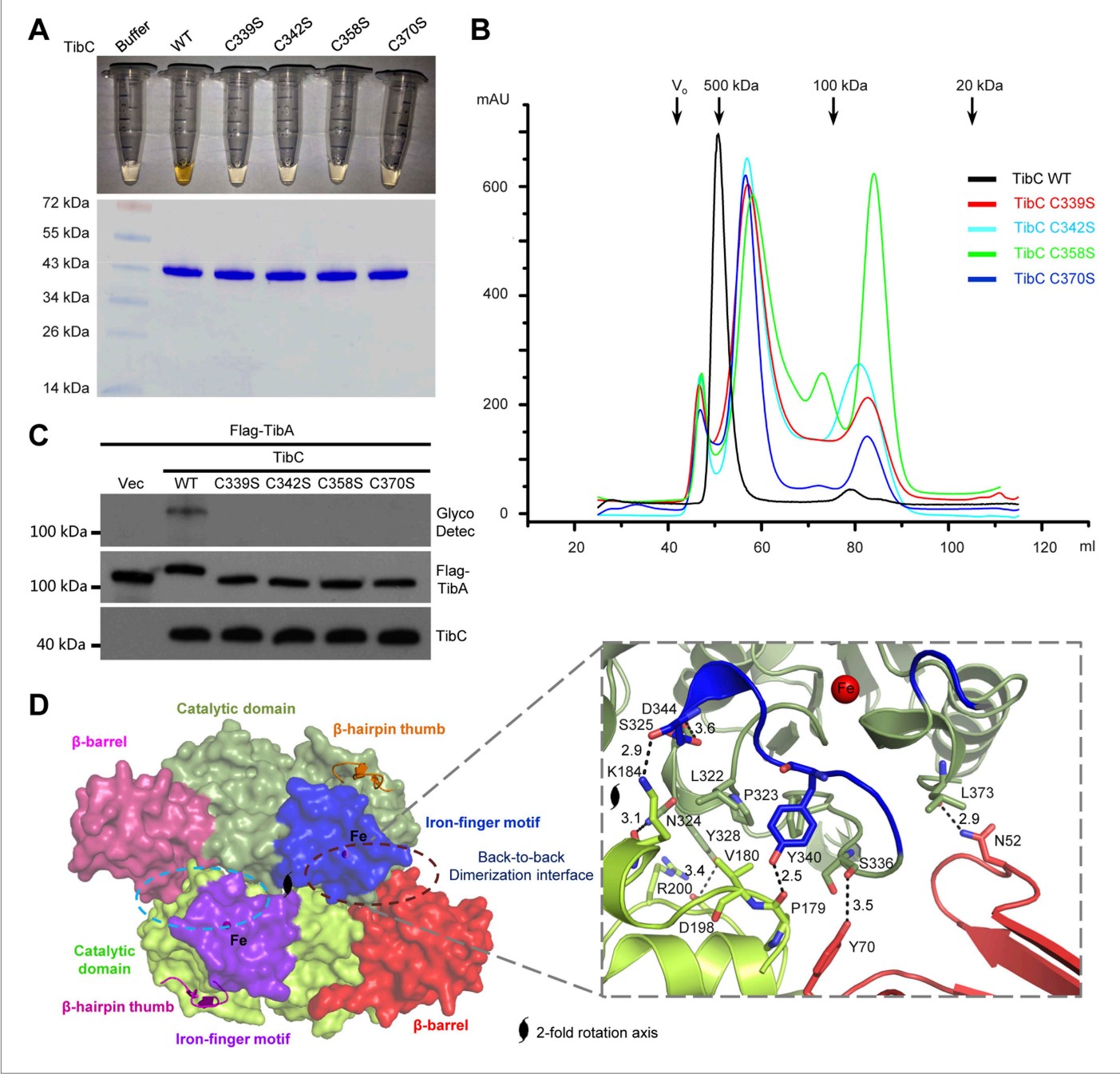

**Figure 5**. The iron-finger motif critical for TibC dodecamer assembly and heptosyltransferase activity. (**A**) Purified TibC (WT and the four iron-finger cysteine mutants, ~8 mg/ml) were loaded onto a SDS-PAGE gel followed by Coomassie blue staining. (**B**) Gel filtration chromatography of the iron-finger cysteine mutants of TibC. Black arrows mark the molecular weight calibration and Vo denotes the void volume. (**C**) Analyses of TibA glycosylation by the iron-finger cysteine mutants of TibC. Indicated proteins were co-expressed in *E. coli* BL21 (DE3) and the lysates were analyzed by anti-Flag (TibA) and anti-TibC immunoblotting and also TibA heptosylation assay. (**D**) The back-to-back dimer formation. Shown on the left is the surface presentation colored as indicated. Two symmetric dimerization interfaces are marked by dashed circles. Structural details of one interface are shown on the right with interacting residues in sticks. Polar interactions are represented by black dashed lines with a number denoting the distance in angstrom.

interface (*Figure 5D*), burying ~1467 Å$^2$ solvent-accessible surface area of TibC (~9% of the total surface areas). Dimerization via this large interface provides building blocks for TibC dodecamer assembly.

## Structure of the TibC dodecamer in complex with ADP-D,D-heptose

We also crystallized the TibC dodecamer in complex with its natural ligand ADP-D-glycero-β-D-manno-heptose (ADP-D,D-heptose). This was accomplished by soaking the sugar ligand into crystals of the catalytically inactive TibC D110A mutant. Structural determination was achieved by molecular replacement, which revealed extra omit density in the TibC catalytic domain. Placing the sugar into the omit density led to a concomitant loss of $F_o$–$F_c$ density after the refinement. ADP-D,D-heptose was convincingly modeled into each protomer and a final model of 3.88 Å was obtained (*Table 1*). The 12 sugar ligands form an array of two parallel circles along the inner surface of the dodecamer ring and face towards the hollow center (*Figure 6A*). ADP-D,D-heptose is located at the cleft between the two lobes of the catalytic domain (*Figure 6B* and *Figure 3B*). The adenine heterocycle fits snugly into a hydrophilic pocket and its N1 atom bears a hydrogen bond contact with Arg-286 of TibC (*Figure 6B*). The hydrogen bond contact, though weak, is incompatible with a guanine due to its saturated N1, providing a structural basis for the specificity of TibC in using ADP-activated sugar donor. Moreover, the heptose hydroxyl O4 forms a hydrogen bond with Trp-305 indolic N1 (*Figure 6B*). The β-phosphate has strong interactions with Thr-226 and Lys-230 in the ligand-binding pocket of TibC. Lys-230 is also close to the heptose hydroxyl O5; its positive charge could potentially alleviate the negative charge of β-phosphate, thereby facilitating nucleophilic attack of the heptose C1 by the substrate acceptor.

Superimposition of the TibC sugar binding site with those of the GT-B glycosyltransferases WaaC and MurG identified Asp-110 as the candidate catalytic base that activates the substrate acceptor. Asp-110 is located close to the anomeric hydroxyl of heptose in the TibC–ADP-D,D-heptose complex structure (*Figure 6B*). All four sugar-binding residues (Arg-286, Trp-305, Thr-226, and Lys-230) as well as the catalytic base (Asp-110) are highly conserved in the BAHT family (*Figure 1—figure supplement 1*). Alanine substitution of Asp-110, Arg-286, Trp-305, or Lys-230 in TibC all abolished or largely diminished TibA/AIDA-I heptosylation in *E. coli* (*Figure 6—figure supplement 1A*); these point mutants also failed to support AIDA-I-mediated bacterial adhesion to HeLa cells (*Figure 6—figure supplement 1B*). The results also confirm the functional importance of the heptosyltransferase activity of TibC and the BAHT family.

## Sugar ligand stereoselectivity

AAH accepts both ADP-D,D-heptose and ADP-L,D-heptose as the sugar donor despite a possible slight preference for the latter (*Figure 6C* and *Figure 6—figure supplement 2*). In contrast, TibC exhibited a high stereoselectivity for ADP-D,D-heptose. To reveal the underlying structural basis, we modeled the structure of sugar ligand-bound AAH using the TibC complex as the template and examined TibC/AAH residues within 5 Å from the sugar ligand (*Figure 6D*). This led to the identification of residue 300 in TibC (294 in AAH) as a possible structural determinant. In TibC, residue 300 is a proline and its main-chain carbonyl oxygen forms two hydrogen bonds with D,D-heptose hydroxyl O4 and O6, respectively (*Figure 6E*). Notably, the latter interaction is incompatible with L,D-heptose due to the chiral inversion of C6 and the resulting spatial distance. For AAH, the side-chain hydroxyl of Ser-294 can have a polar interaction with hydroxyl O6 in D,D-heptose or hydroxyl O7 in L,D-heptose (*Figure 6E*), thereby capable of tolerating both sugar ligands. Consistent with these structural indications, TibC P300S mutant could use both ADP-D,D-heptose and ADP-L,D-heptose while AAH S294P mutant instead became selective for ADP-D,D-heptose (*Figure 6C* and *Figure 6—figure supplement 2A*). Thus, Pro-300-mediated polar interaction is a critical structural determinant for TibC sugar ligand stereoselectivity.

## Cryo-EM analysis of TibC–TibA complex reveals two conformational states

We further succeeded in purifying a TibC–TibA enzyme/substrate complex by co-expression of the sugar binding-deficient TibC K230A mutant with TibA$_{55-350}$ in *E. coli*. The complex eluted from the gel filtration column as a homogenous fraction with a size larger than the TibC dodecamer. The molecular weight of the giant enzyme/substrate assembly was determined to be 750 kDa by analytical ultracentrifugation (*Figure 2—figure supplement 2*), matching well the composition of a TibC dodecamer and six TibA molecules. Cryo-EM micrographs revealed a well-defined shape of the TibC–TibA dodecamer/hexamer complex at an intermediate resolution (*Figure 7—figure supplement 1A*), allowing for single particle structural analysis (*Figure 7—figure supplements 1, 2*). Averaging and refinement without classification from the whole dataset of 53,303 particles produced an electron density map of

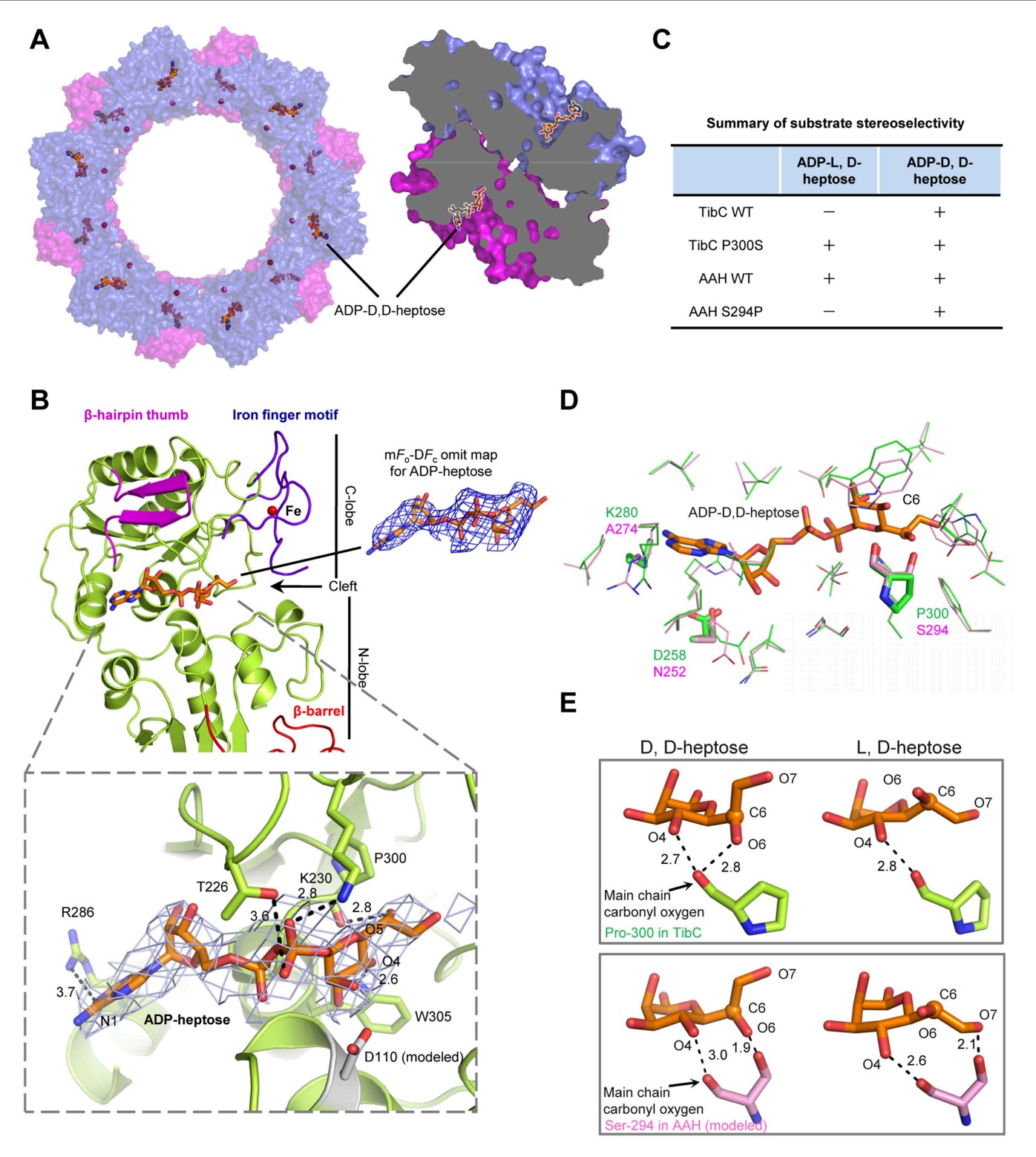

**Figure 6**. Crystal structure of the TibC-ADP-D,D-heptose complex and sugar ligand stereoselectivity. (**A**) Structure of the TibC-ADP-D,D-heptose complex. Left, a top view of the dodecamer. Right, a half-cut view of the back-to-back dimer. ADP-D,D-heptose and ferric ions are shown as ball-and-stick models and red spheres, respectively. (**B**) Binding of ADP-D,D-heptose to TibC protomer. Key sugar-binding residues are in sticks. Polar interactions are indicated by black dashed lines with a number denoting the distance in angstrom. ADP-D,D-heptose is shown as orange sticks meshed with σA-weighted $2mF_o$-$DF_c$ electron density map contoured at 1.0 σ. The $mF_o$–$DF_c$ omit density of the ligand at a contour level of 3.0 σ is shown in blue on *Figure 6. Continued on next page*

*Figure 6. Continued*

the right. (**C**) Sugar ligand stereoselectivity and effects of swapping Pro-300 in TibC with the corresponding Ser-294 in autotransporter adhesin heptosyl-transferase (AAH). The raw experimental data are in *Figure 6—figure supplement 2A*. (**D**) Comparison of ADP-D,D-heptose binding residues in TibC (green) and AAH (pink). The AAH structure was modeled from that of TibC. Conserved residues are shown as lines; divergent residues are in sticks. ADP-D,D-heptose is in orange sticks with the chiral C6 as a sphere. (**E**) A proposed model for the ligand stereoselectivity. The stereoselectivity determinants (Pro-300 in TibC and Ser-294 in AAH) are in sticks. The distance between the sugar ligand and the Pro-300/Ser-294 are denoted as a number in angstrom.

The following figure supplements are available for figure 6:

**Figure supplement 1**. Requirement of TibC heptosyltransferase activity for bacterial adhesion to host cells.

**Figure supplement 2**. Assays of the ligand stereoselectivity of TibC and autotransporter adhesion heptosyltransferase (AAH).

9.7 Å in which the TibC$_{12}$–TibA$_6$ stoichiometry became immediately evident (*Figure 7A*). In this low-resolution model, density agreeing well with the crystal structure of the TibC dodecamer could be clearly identified, but the TibA moiety is slightly blurred and segmented, indicating a conformational heterogeneity. The reconstructed maps could be further classified into two classes based on different features of the TibA density (*Figure 7—figure supplement 2*). Two maps of higher resolution at 8.2 Å and 8.9 Å were then reconstructed by averaging 35,300 and 18,003 sorted particles, respectively (*Figure 7—figure supplement 2*). As a result, a higher contrast in the TibA structure was observed between the two models despite a similar overall architecture for the entire complex (*Figure 7A*). These analyses indicate two major conformational states of the TibC–TibA enzyme/substrate complex in the absence of the sugar donor.

## Resting state of the TibC–TibA dodecamer/hexamer complex

The 8.9 Å EM map fits remarkably well with crystal structures of TibA$_{55-350}$ (PDB ID code: 4Q1Q) (*Lu et al., 2014*) and the TibC dodecamer (*Figure 7B*), allowing for reliable atomic model docking. The cross-correlation coefficient (CCC) for TibA in the final model is 0.76 (only a small N-terminal portion of TibA$_{55-350}$ is out of the density). The reconstruction revealed that six TibA$_{55-350}$ molecules are distributed along the inner surface of the TibC dodecamer ring and share its sixfold axis (*Figure 7C*). Importantly, each TibA$_{55-350}$ binds to two TibC protomers of a back-to-back dimer, suggesting that two enzyme molecules are responsible for catalyzing full heptosylation of one TibA substrate. Ser-176 is the closest sugar acceptor situated at the entrance of the catalytic cleft of one TibC protomer (TibC-A), but the distance to the catalytic base exceeds 20 Å (*Figure 7C* and *Figure 7—figure supplement 3A*). A similar distance was observed between the other TibC protomer (TibC-B) and its possible substrate acceptor (*Figure 7—figure supplement 3A*). Thus, the 8.9 Å TibC–TibA dodecamer/hexamer complex model represents a sugar transfer incompetent state, designated hereafter as the resting state. Moreover, the spiral shaft of TibA β-helix adopts a trend roughly parallel to the sixfold axis of the dodecamer which, together with the patterned solenoid-like surface distribution of heptosylation sites on TibA (*Lu et al., 2014*), indicates a possible screw propelling-like mechanism for processive heptosylation of TibA by the dodecamer ring.

## Active state of the TibC–TibA dodecamer/hexamer complex

The static X-ray structures of TibC dodecamer and TibA$_{55-350}$ were also docked into the 8.2 Å EM map, but the TibA moiety had a few degrees of bending with a protrusion into the catalytic center of one TibC protomer (*Figure 7A,B* and *Figure 7—figure supplement 3B*). The extra protruding density could not be appropriately fit despite extensive attempts at possible solutions. The heterogeneity observed in the cryo-EM image likely reflects the dynamic catalytic motions; the protruding density was tentatively interpreted as the TibA loop bearing the sugar acceptor Ser-176. This interpretation was supported by molecular dynamics (MD) simulation of a resting state TibC$_2$–TibA$_{55-350}$ trimer. In the 400 ps steered MD simulation, pulling Ser-176 in TibA$_{55-350}$ towards the heptose led to a proportional increase in the CCC (*Figure 7D, E*). A subsequent 20 ns unbiased MD simulation produced an ensemble of TibA models that matched the EM density with a best overall CCC value of 0.87 (*Figure 7B*). The final model of the entire complex was well accommodated by the experimental map (*Figure 7B* and *Figure 7—figure supplement 3B*). The protruding loop was assigned to residues

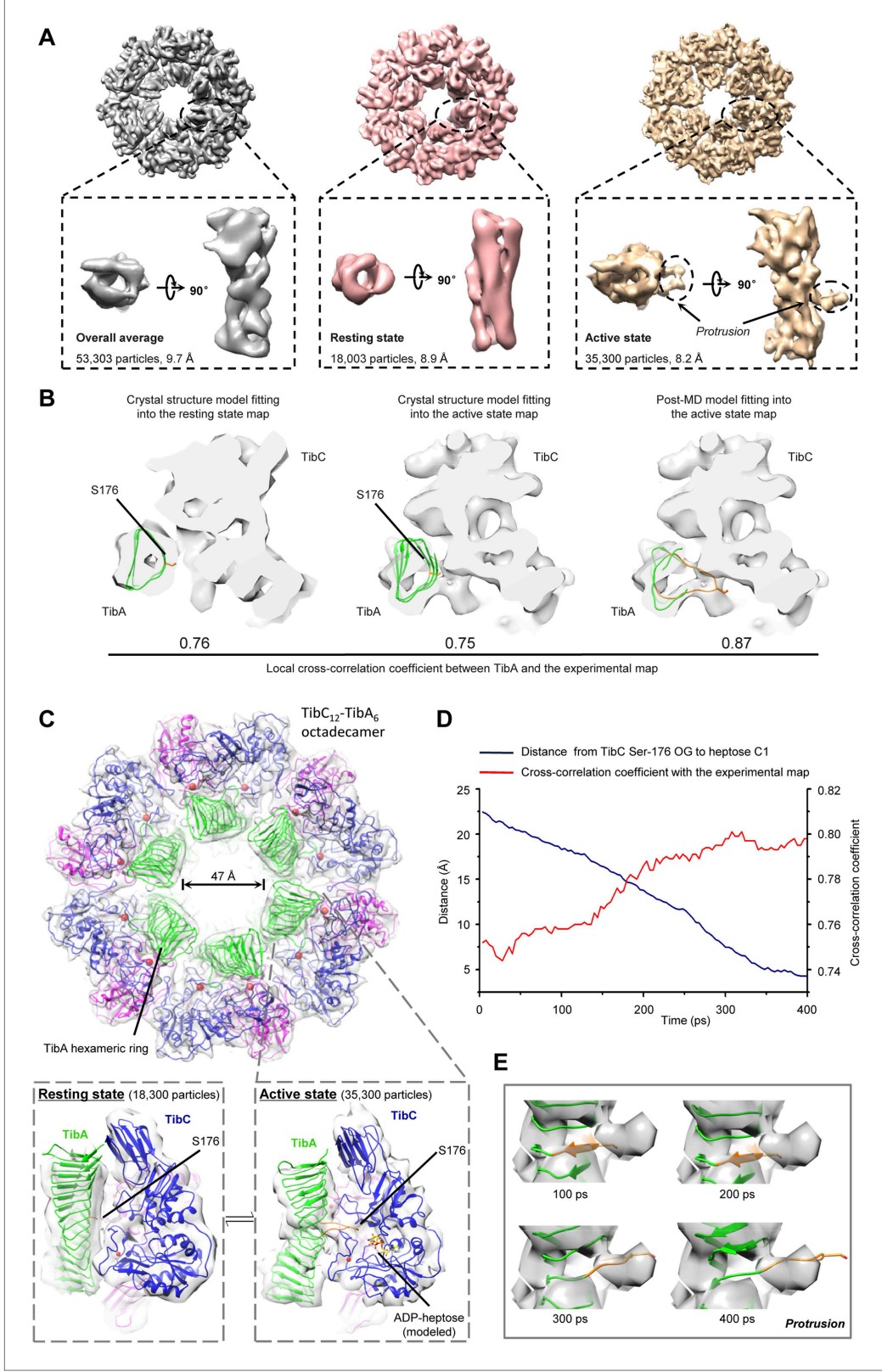

**Figure 7**. Cryo-EM models of the TibC–TibA$_{55-350}$ dodecamer/hexamer complex. (**A**) Cryo-EM maps of the TibC–TibA$_{55-350}$ complex. The left is the overall average map obtained from classification-free reconstruction. A two-class classification was employed to distinguish different features observed with the TibA density

*Figure 7. Continued on next page*

*Figure 7. Continued*

(*Figure 7—figure supplement 2*), and the resulting two maps are shown in the middle and right panels. (**B**) Model fitting of TibA$_{55-350}$ crystal structure into the two classes of experimental maps and comparison with the post-molecular dynamics (MD) model. The cross-correlation coefficient (CCC) between TibA and the experimental map is listed underneath the models. The protrusion loop is highlighted in orange and the to-be-glycosylated Ser-176 is in sticks. (**C**) Cryo-EM reconstruction of the TibC–TibA dodecamer/hexamer complex. The atomic model after MD simulation was fit into the cryo-EM map (the upper panel). Ribbon diagram of TibC dodecamer is colored as that in *Figure 2A*. The lower panel shows the TibA–TibC$_2$ trimer in the active and resting states. ADP-heptose and Ser-176 are in sticks. The conformationally changed TibA loop is highlighted in orange. (**D**) The CCC for TibA during the steered MD simulation. The distance from TibA Ser-176 OG to heptose C1 (blue line) and the local CCC value for TibA (red line) are plotted as a function of time. (**E**) Protrusion of the TibA loop sampled by the steered MD simulation. The snapshots at 100, 200, 300, and 400 ps were fitted into the active state cryo-EM map. The protrusion loop is in orange and Ser-176 is in sticks.

The following figure supplements are available for figure 7:

**Figure supplement 1**. Cryo-EM images of the TibC-TibA$_{55-350}$ complex.

**Figure supplement 2**. Cryo-EM reconstruction of the TibC$_{12}$-TibA$_6$ complex.

**Figure supplement 3**. Molecular dynamics (MD)-assisted cryo-EM analysis reveals two conformational states of the TibC–TibA complex.

170–180 of TibA, resulting from disruption of a local β-strand within these residues. The loop placed Ser-176 into a position ready for attacking the sugar donor in one TibC protomer (TibC-A) (*Figure 7C* and *Figure 7—figure supplement 3C*). This 'protrusion' state of TibA was designated as the catalytically competent active state. Notably, the other TibC protomer (TibC-B) in the back-to-back dimer and its possible substrate acceptor appeared to be in the resting state (*Figure 7—figure supplement 3C,D*), suggesting that only one of the two enzyme molecules in the dimer can be productive in transferring the sugar onto TibA at a time. It is worth mentioning here that the EM dataset presumably covers a continuous series of catalytic stages; the two discrete classifications employed in our reconstruction may not give a complete picture of all catalytic motions of the TibC–TibA enzyme/substrate complex. The active state structure, which was reconstructed from two-thirds of the total particles, probably contains the majority of representative conformational stages in the catalytic process and therefore is used to represent the overall structure of the TibC–TibA dodecamer/hexamer complex (*Figure 7C*).

## Discussion

### The large ring-shape platform of TibC dodecamer for efficient heptosylation of TibA

TibC is distinct from known glycosyltransferases in that it adopts an unusual giant dodecamer ring architecture in catalyzing simultaneous glycosylation of six molecules of TibA autotransporter. The selection of a double-layered ring with a large hollow channel is advantageous and provides a favorable environment for cooperative catalysis in a structurally economical manner. Assembly of the dodecamer into a ring can reduce the solvent-accessible surface area and renders a stable enzymatic machinery complex beneficial to efficient catalysis. Moreover, the spatial arrangement of the back-to-back TibC dimer, binding to the different glycosylation sites in one TibA, suggests a divalent catalysis is also probably contributing to the high catalytic efficiency required for hyperglycosylation. Thus, the dodecamer/hexamer enzyme/substrate complex with six symmetric units may represent a best compromise between structural economy and catalytic cooperativity/efficiency.

The TibC complex, compared with other macromolecular assemblies with large interior volumes such as ferritin and multimeric chaperones/proteases, is more permeable with 12 titled grooves extending from outside to the interior chamber. These grooves are optimally imbedded and close to the catalytic center, allowing for easy access of ADP-heptose from outside into the TibC particle. The catalytic center, on the other hand, is structurally best accessible from the interior, where TibA is

located and can readily deliver the to-be-glycosylated serine by loop protrusion. We further speculate that the energy used to drive the rotation and shuttle motion of the TibA substrate is partially derived from breakdown of the ADP-heptose glycosidic bond occurring during catalysis. Further experiments such as obtaining a higher resolution cryo-EM structure of the $TibC_{12}$–$TibA_6$ complex, which may capture more catalytic stages/events, are necessary to validate the proposed model for TibC-catalyzed processive heptosylation of TibA.

### The unique iron-finger motif

Another most unusual feature of the TibC protomer structure is the presence of the 1Fe-0S iron-finger motif. The iron-finger motif does not fulfill the previously defined function of serving as an electron carrier in rubredoxin protein. Biochemical and structural studies have underscored the singular importance of this motif in mediating the back-to-back dimer formation and TibA substrate recognition. Interestingly, the iron-finger motif, adopting a topological fold similar to the zinc-finger motif, follows a right-handed helical path in binding to TibA β-helix, which is analogous to DNA binding by zinc-finger motifs. In addition, the recognition specificity and the interaction strength of this motif will have been precisely tuned during macromolecular evolution so that specific substrate recognition, sliding of the TibA β-helix along the reaction channel, and efficient protrusion of the TibA loop can all occur in a catalytically productive manner. It will be interesting to elucidate the detailed mechanism by which iron-finger motif-mediated substrate recognition is effectively coupled with its promoting of the TibA loop protrusion.

## Materials and methods

### Constructs and recombinant proteins

The cDNAs encoding TibC and TibA were PCR amplified from genomic DNA of ETEC strain H10407. The full-length *tibC* was cloned into pGEX-6p-2 vector (GE Healthcare, UK) for recombinant expression in *E. coli*. To co-express $TibA_{55-350}$ and TibC K230A for capturing the TibC–$TibA_{55-350}$ enzyme–substrate complex, the *tibA* fragment was cloned into the pGEX-6p-2 vector and a catalytically inactive mutant of TibC was inserted into the pACYCDuet vector. AAH and AIDA-I constructs have been described previously (*Lu et al., 2014*). All constructs were generated by standard molecular cloning. Point mutations were generated by the QuickChang Site-Directed Mutagenesis Kit (Stratagene, La Jolla, CA). All the plasmids were verified by DNA sequencing.

All recombinant proteins and protein complexes were expressed in *E. coli* BL21 (DE3) Gold strain (Agilent Technologies, Santa Clara, CA) with 0.5 mM isopropyl β-D-1-thiogalactopyranoside (IPTG) for overnight induction at 22°C in Luria–Bertani (LB) medium. TibC was purified by glutathione affinity chromatography and the GST tag was removed by PreScission protease digestion. TibC was further purified by anion exchange and gel filtration chromatography using HiTrap Q HP and Superdex 200 Hi-load columns (GE Healthcare), respectively. The TibC–$TibA_{55-350}$ complex was initially purified using one-step glutathione affinity chromatography. The GST moiety on TibA was removed by PreScission protease cleavage. The complex was further separated from free $TibA_{55-350}$ by gel filtration chromatography using the Superdex 200 Hi-Load column. SeMet-labeled TibC and TibA were expressed in the methionine-auxotrophic *E. coli* B834 (DE3) strain (Novagen, Germany) using SeMet-supplemented M9 medium. SeMet protein was purified following the same scheme as that used for the native protein.

### Crystallization and data collection

All the crystallization experiments were carried out at 20°C using the hanging drop vapor diffusion method. The purified TibC was concentrated to 20 mg/ml in a buffer containing 10 mM Tris–HCl (pH 7.6), 100 mM NaCl, and 2 mM DTT. Crystals of both native and SeMet-TibC protein were grown in 8% (wt/vol) PEG 8000, 120 mM magnesium acetate, and 100 mM MES buffer (pH 5.5) for two days. Crystals were cryoprotected with the mother liquid containing 19% (vol/vol) ethylene glycol and 1% (vol/vol) DMSO and flash-frozen with liquid nitrogen for data collection. The TibC D110A mutant crystals were grown at the same condition. Before freezing, the crystals were soaked with 5 mM ADP-D,D-heptose in the mother liquid for 1 hr.

### Structure determination

All the diffraction data were collected at BL-17U of Shanghai Synchrotron Radiation Facility (SSRF) at 100 K and processed with the HKL 2000 suite by the routine procedure (*Otwinowski and Minor, 1997*).

The structure of TibC dodecamer was solved by the single wavelength anomalous dispersion (SAD) method. In the TibC data, a total of 36 heavy atoms were determined using the program SHELXD (*Schneider and Sheldrick, 2002*). The identified heavy atom sites were then refined and the initial phases were calculated using the program autoSHARP (Global Phasing Ltd). The phase was then refined and extended to 2.9 Å by NCS averaging, solvent flattening and histogram matching using the DM module in CCP4 (*Dodson et al., 1997*). After density modification, the map was sufficiently interpretable for model building. The model was automatically built using the Buccaneer software (*Cowtan, 2006*), which generated ~70% complete model. The remaining parts were manually built and adjusted in Coot (*Emsley et al., 2010*). The model was refined in Refmac (*Murshudov et al., 1997*). The TibC–ADP-D,D-heptose complex structure was solved by molecular replacement in Phaser MR (*McCoy et al., 2007*) using the apo-TibC structure as the template. The final model was adjusted in Coot and refined in Refmac. The structural pictures were generated in Pymol (www.pymol.org).

## Gel filtration chromatography analysis of the protein size

Purified recombinant TibC (WT or indicated mutants) were loaded onto a HiLoad 16/600 Superdex 200 column (GE Healthcare) in a buffer containing 20 mM Tris–HCl (pH 7.6) and 20 mM NaCl on an ÄKTA Purifier (GE Healthcare). Fractions corresponding to the specified elution volumes were collected and loaded onto SDS-PAGE gels followed by Coomassie blue staining.

## In vivo and in vitro heptosyltransferase enzymatic assays

In vivo heptosyltransferase activity of TibC was monitored by co-expressing pACYCDuet-Flag-TibA (the Flag tag was inserted at the junction between the signal peptide and the passenger domain) and pET21a-TibC (WT and indicated mutants). 1/10 of 100 µl of total lysates from 1 mL of *E. coli* culture was used for immunoblotting and glycoprotein detection analyses. Expression levels of TibA and TibC were detected by anti-Flag M2 mouse monoclonal antibody (Sigma-Aldrich, St. Louis, MO) and in-house generated rabbit anti-TibC serum, respectively. Glycosylated proteins were detected with the ECL glycoprotein detection module (GE Healthcare, product code RPN2190) according to the manufacturer's protocol. For in vitro analysis of TibC and AAH heptosyltransferase activities, 10 µg of the TibA-derived synthetic peptide (GGVQHVSSGGSATSSTINSGG, synthesized by Scilight Biotechnology LLC, China) were incubated with 2 µg of purified TibC or AAH protein (WT or indicated mutants) for 4 hr at 30°C in a 20 µl reaction containing 10 mM Tris–HCl (pH 7.5), 50 mM NaCl, 10 mM $MgCl_2$, 1 mM DTT, and 0.5 mM ADP-L,D-heptose or ADP-D,D-heptose ligands (*Zamyatina et al., 2003*). The reaction mixtures were subjected to mass spectrometry analysis.

## Analytic ultracentrifugation

Analytic ultracentrifugation was carried out with an XL-I analytical ultracentrifuge (Beckman-Coulter, Indianapolis, IN) with An-60 Tirotor at 4°C. The TibC and TibC–TibA complex proteins were prepared at 0.8 mg/ml in a buffer containing 20 mM Tris–HCl (pH 8.0), 150 mM NaCl, and 2 mM DTT. All data were collected at a speed of 25,000/32,000 rpm. 3 mM path length charcoal-filled Epon centerpieces were used for the TibC–TibA complex due to the sample quality limitation and a 12 mM path length aluminum centerpiece was used for the TibC sample. Samples were monitored real time using interference optics at intervals of 4 min. Buffer alone was used as the reference. The molecular weights were calculated by the SEDFIT program (*Schuck, 2000*).

## Cryo-EM data acquisition, image processing, and reconstruction

10 µg/ml bacitracin (Sigma) was added to the purified protein to obtain mono-dispersed particles and make the orientation distribution more anisotropic. An aliquot of 3.5 µl of TibC or TibC–TibA$_{55-350}$ complex (~1 mg/ml) was absorbed onto a glow-discharged Quantifoil holey carbon grid (R2.1, 300 mesh; Jena Biosciences, Germany). The grid was blotted in an FEI Vitrobot Mark IV (FEI Company, Hilsboro, OR) using 4 s blotting time and blotting force 2 with 100% humidity at 283 K, and then plunged into the chilled liquid ethane. The grids were immediately transferred into a Titan Krios electron microscope (FEI Company) operated at 300 kV. For the TibC dodecamer, approximately 3000 images were semi-automatically collected following the Leginon procedure (*Carragher et al., 2000*) using a 4 K × 4 K Gatan UltraScan4000 CCD (model 895, Gatan Inc, Pleasanton, CA) at a nominal magnification of 75,000×. For the TibC–TibA$_{55-350}$ complex, ~1800 cryo-EM images were recorded using a 2K × 2K Gatan Tridiem CCD camera (Gatan Inc) at a nominal magnification of 81,000× with the energy filter turned off. The corresponding pixel size on the specimen level is 1.196 and 1.778 Å/pixel for TibC and

TibC–TibA$_{55-350}$ complex datasets, respectively. The range of nominal defocus is 2.3–3.6 µm for TibC and 2.7–4.1 µm for TibC–TibA$_{55-350}$ data. The electron dose is 20 e−/Å·s for each CCD frame. For both TibC and TibC–TibA$_{55-350}$ micrographs, the particles were semi-automatically boxed using e2boxer.py in EMAN2 package (*Tang et al., 2007*) followed by interactively screening. The particles with the wrong shape and size were discarded. Approximately 10,200 and 56,000 particle images were cropped out and normalized (mean = 0, standard deviation = 1) for TibC and TibC–TibA$_{55-350}$ complex, respectively. After determining the defocus and B-factor, the particles were phase-flipped for contrast transfer function (CTF) correction using e2ctf.py in EMAN2 (*Tang et al., 2007*).

For the TibC dataset, two-dimensional multivariate statistical analysis-based reference-free classification was performed in EMAN2 to further eliminate particles belonging to the class averages with blurry appearance. Particles contained in the averages, which were redundant in the ring-like view, were also removed to avoid introducing orientation isotropy. A total of 8546 particles from the TibC data were used for subsequent data processing. The initial model was built using e2initialmodel.py with C1 symmetry applied and was iteratively refined against the phase-flipped particles until no further improvement could be obtained. During the projection-matching refinement process, a Fourier ring correlation (FRC) was chosen to compare the similarity between the class averages and re-projections of the model, and C6 symmetry was enforced as indicated by the reference-free class averages. 20% of the worst particles in each class were discarded in each round of iteration. The information over a 20–100 Å resolution range and SSNR weighting were considered in the comparison in several early iterations. In the last four rounds of iterations, the information was expanded to the full resolution range and the step angle was assigned to 2.0°. The map obtained in the last iteration was then subjected to B-factor correction with a damping factor of −210 Å$^2$ to enhance the high resolution information using the program embfactor (*Fernandez et al., 2008*).

For the TibC–TibA$_{55-350}$ complex data, the processing procedure before reconstruction was similar to that described for the TibC data above. A total of 53,303 particles were finally employed for 3D reconstruction. The starting model used was a 60 Å low-pass filtered TibC dodecamer reconstruction (C6 symmetry applied). After several rounds of refinement running, the phase-flipped particle images were imported into RELION (*Scheres, 2012*) for 3D classification. After 30 rounds of iterations with three-class classification, two classes that contained 35,300 members in total and showed almost identical reconstruction in the TibA density were merged together for subsequent 3D refinement. Finally, two clusters that corresponded to the 'active' and 'resting' conformations of the complex were acquired. After the 3D refinement converged, the particles corresponding to the two clusters were sorted for subsequent single-model projection-matching refinement in EMAN2. Final maps for the two conformations were independently reconstructed through projection-matching refinement with the step angle of 1.68° for the active conformation and 3.6° for the resting conformation. Resolution was estimated by the gold standard FSC using the 0.143 cutoff value (*Rosenthal and Henderson, 2003*; *Scheres and Chen, 2012*). An additional projection-matching reconstruction test for potential model bias was performed with the resting volume as the initial reference against the active state particle set and vice versa. The structural pictures of cryo-EM reconstruction were drawn in UCSF chimera (*Pettersen et al., 2004*).

## Molecular modeling of AAH structure

Homology modeling of AAH was performed in the program MODELLER (version 9v7) (*Marti-Renom et al., 2000*; *Bredenberg and Nilsson, 2001*) using the TibC structure as the template. The quality of the modeled structure was ensured by the high sequence identity between AAH and TibC (68%) (*Figure 1—figure supplement 1*). The top DPOE scored model was chosen and validated with PROCHECK (*Laskowski et al., 1993*). Structural models of AAH S294P and TibC P300S mutants were generated directly from the corresponding wild-type structures. Residues within a radius of 5 Å from the mutated residue were refined using the Protein Local Optimization Program (PLOP) (*Jacobson et al., 2002*; *Zhu et al., 2007*).

## Refinement of TibC$_{12}$–TibA$_6$ complex structures

Molecular dynamics flexible fitting (MDFF) (*Trabuco et al., 2009*) was used to refine the complex structure by incorporating the EM density map as an external potential into MD sampling. All MD simulations were performed in the program NAMD 2.9 (*Phillips et al., 2005*) using the CHARMM27 force field (*MacKerell et al., 1998*) including the CMAP correction (*Mackerell et al., 2004*). The MDFF simulations were carried out at T = 300 K with a scaling factor ζ = 1 kcal for 10 ns. The snapshot with

the top CCC was selected for further unbiased MD simulation. The value of CCC was calculated using the program VMD (version 1.9.1) (*Humphrey et al., 1996*).

## Molecular dynamics simulation

The CHARMM force field files (topology and parameter) for ADP-D,D-heptose were automatically generated using the server SwissParam (*Zoete et al., 2011*), except that the partial atomic charges of ADP-D,D-heptose were manually assigned using the analogs in the existing CHARMM force field (*Best et al., 2012*). The parameters of ferric ions with four coordinating cysteine residues were modified according to the available parameters developed for the zinc-finger motif (*Eriksson et al., 1995*; *Bredenberg and Nilsson, 2001*).

To sample the intermediate state of loop protrusion in the TibC–TibA complex, steered molecular dynamics (SMD) simulation (*Isralewitz et al., 2001*) was carried out with one unit of $TibC_{12}$–$TibA_6$ complex. By fixing the position of the C1 atom of ADP-D,D-heptose, a force was applied to the OG atom of Ser-176 along a vector connecting the two atoms. The constant pulling velocity was set to 0.06 Å/ps with a spring constant of 10 kcal/mol/Å$^2$. The trajectories of TibA were extracted from a total of 400 ps SMD simulation and then fitted into the cryo-EM density of TibA, which was extracted from the total map using 'Volume Eraser' in CHIMERA (*Pettersen et al., 2004*). The correlation between the CCC values and the distances of loop protrusion was analyzed. The last snapshot of the TibC–TibA subunit from SMD with minimum distance between C1 and OG was chosen as the initial model of the active state conformation, and a 20 ns unbiased MD simulation was then carried out for further refinement. The MD snapshot with the top CCC score was used to generate the final active state TibC–TibA dodecamer/hexamer complex structure.

The MDFF-refined $TibC_{12}$–$TibA_6$ complex structure was solvated using TIP3P water molecules and then neutralized. The system was minimized for 10,000 integration steps and equilibrated for 100 ps with a time step of 1 fs, and the temperature was gradually increased from 25 K to 300 K. Following this, an unbiased MD simulation was performed under a constant temperature of 300 K and a constant pressure of 1 atm using the Nosé–Hoover Langevin piston method (*Feller et al., 1995*) with the integration time step of 2 fs. The cutoff distances for electrostatic and van der Waals calculations were set at 10 Å. The long-range electrostatic forces were computed using the particle mesh Ewald method (*Darden et al., 1993*) with a grid spacing of 1 Å. All covalent bonds involving hydrogen atoms were constrained with the SHAKE algorithm.

## Accession numbers

Crystal structural data for apo-TibC and TibC/ADP-heptose complex are deposited in the Protein Data Bank (PDB) under the accession numbers 4RAP and 4RB4, respectively. Cryo-EM structure data are deposited in Electron Microscopy Data Bank (EMD) with accession codes EMD-2755 (apo-TibC), EMD-2756 (averaged $TibC_{12}$–$TibA_6$ complex), EMD-2757 (active state $TibC_{12}$–$TibA_6$ complex), and EMD-2758 (resting state $TibC_{12}$–$TibA_6$ complex).

## Acknowledgements

We thank the staff at Shanghai Synchrotron Radiation Facility (SSRF) for their assistance in data collection. We thank W Chu at Tsinghua University and the China National Center for Protein Sciences Beijing for performing analytical ultracentrifugation. We thank Dr. Rong Meng and the mass spectrometry facility of National Center for Protein Sciences at Peking University for assistance with ETD mass spectrometry analyses. Computational support was provided by the National Supercomputer Center in Tianjin, and the calculations were performed on TianHe-1(A). We are also grateful to all members of the Shao laboratory for discussions and technical assistance. The research was supported in part by an International Early Career Scientist grant from the Howard Hughes Medical Institute to FS. This work was also supported by the National Basic Research Program of China 973 Programs (2012CB518700 and 2014CB849602), the Strategic Priority Research Program of the Chinese Academy of Sciences (XDB08020202) to FS, and the China National Science Foundation Program for Distinguished Young Scholars to FS (31225002) and PZ (31230018).

## Additional information

### Competing interests

FS: Reviewing editor, *eLife*. The other authors declare that no competing interests exist.

## Funding

| Funder | Grant reference number | Author |
|---|---|---|
| Howard Hughes Medical Institute | 55007431 | Feng Shao |
| Ministry of Science and Technology of the People's Republic of China | 2012CB518700 and 2014CB849602 | Qing Yao, Qiuhe Lu, Yue Xu, Feng Shao |
| National Natural Science Foundation of China | Program for Distinguished Young Scholars 31225002 | Feng Shao |
| Chinese Academy of Sciences | XDB08020202 | Feng Shao |
| National Natural Science Foundation of China | Program for Distinguished Young Scholars 31230018 | Ping Zhu |

The funders had no role in study design, data collection and interpretation, or the decision to submit the work for publication.

## Author contributions

QY, QL, Conception and design, Acquisition of data, Analysis and interpretation of data, Drafting or revising the article; XW, Acquisition of data, Analysis and interpretation of data, Drafting or revising the article; FS, MH, XL, Acquisition of data, Analysis and interpretation of data; YX, Conception and design, Acquisition of data, Analysis and interpretation of data; AZ, Analysis and interpretation of data, Contributed unpublished essential data or reagents; NH, Conception and design, Analysis and interpretation of data; PZ, FS, Conception and design, Analysis and interpretation of data, Drafting or revising the article

# Additional files

## Major datasets

The following datasets were generated:

| Author(s) | Year | Dataset title | Dataset ID and/or URL | Database, license, and accessibility information |
|---|---|---|---|---|
| Yao Q, Lu Q, Xu Y, Shao F | 2014 | Crystal structure of bacterial iron-containing dodecameric glycosyltransferase TibC from enterotoxigenic E. coli H10407 | http://www.pdb.org/pdb/search/structidSearch.do?structureId=4RAP | Publicly available at RCSB Protein Data Bank. |
| Yao Q, Lu Q, Shao F | 2014 | Crystal structure of dodecameric iron-containing heptosyltransferase TibC in complex with ADP-D-beta-D-heptose at 3.9 angstrom resolution | http://www.pdb.org/pdb/search/structidSearch.do?structureId=4RB4 | Publicly available at RCSB Protein Data Bank. |
| Yao Q, Lu QH, Wan XB, Song F, Xu Y, Zamyatina A, Huang N, Zhu P, Shao F | 2014 | Cryo-electron microscopy of TibC dodecamer | http://www.ebi.ac.uk/pdbe/entry/EMD-2755 | Publicly available at Electron Microscopy Data Bank. |
| Yao Q, Lu QH, Wan XB, Song F, Xu Y, Zamyatina A, Huang N, Zhu P, Shao F | 2014 | Cryo-electron microscopy of TibC12-TibA6 octadecamer in averaged conformation | http://www.ebi.ac.uk/pdbe/entry/EMD-2756 | Publicly available at Electron Microscopy Data Bank. |
| Yao Q, Lu QH, Wan XB, Song F, Xu Y, Zamyatina A, Huang N, Zhu P, Shao F | 2014 | Cryo-electron microscopy of TibC12-TibA6 octadecamer in active state | http://www.ebi.ac.uk/pdbe/entry/EMD-2757 | Publicly available at Electron Microscopy Data Bank. |
| Yao Q, Lu QH, Wan XB, Song F, Xu Y, Zamyatina A, Huang N, Zhu P, Shao F | 2014 | Cryo-electron microscopy of TibC12-TibA6 octadecamer in resting state | http://www.ebi.ac.uk/pdbe/entry/EMD-2758 | Publicly available at Electron Microscopy Data Bank. |

The following previously published datasets were used:

| Author(s) | Year | Dataset title | Dataset ID and/or URL | Database, license, and accessibility information |
|-----------|------|---------------|----------------------|------------------------------------------------|
| Lu Q, Yao Q, Xu Y, Li L, Li S, Liu Y, Gao W, Niu M, Sharon M, Ben-Nissan G, Zamyatina A, Liu X, Chen S, Shaoemail F | 2014 | Crystal structure of TibC-catalyzed hyper-glycosylated TibA55-350 fragment | http://www.pdb.org/pdb/explore/explore.do?structureId=4Q1Q | Publicly available at RCSB Protein Data Bank. |
| Grizot S, Salem M, Vongsouthi V, Durand L, Moreau F, Dohi H, Vincent S, Escaich S, Ducruix A | 2006 | E. coli heptosyltransferase WaaC with ADP-2-deoxy-2-fluoro heptose | http://www.pdb.org/pdb/explore/explore.do?structureId=2H1H | Publicly available at RCSB Protein Data Bank. |
| Hu Y, Chen L, Ha S, Gross B, Falcone B, Walker D, Mokhtarzadeh M, Walker S | 2003 | Crystal structure of MurG:GlcNAc complex | http://www.pdb.org/pdb/explore/explore.do?structureId=1NLM | Publicly available at RCSB Protein Data Bank. |

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
