## [Decision Letter]

Thank you for sending your work entitled “A structural mechanism for bacterial autotransporter glycosylation by a dodecameric heptosyltransferase family” for consideration at *eLife*. Your article has been favorably evaluated by Richard Losick (Senior editor) and 3 reviewers, one of whom is a member of our Board of Reviewing Editors.

The Reviewing editor and the other reviewers discussed their comments before we reached this decision, and the Reviewing editor has assembled the following comments to help you prepare a revised submission.

The authors show, both in *E. coli* and *in vitro*, that TibC heptosylates the autotransporters TibA and AIDA-I. These findings are both notable for the identification of the enzyme responsible for glycosylation, because the sequence did not reveal such activity, and because of the dearth of information about heptosylating enzymes. In addition, the authors provide the X-ray structure of TibC, the first of the BAHT family of bacterial heptosyltransferases. The structure is an unusual ring-shaped dodecamer, with a catalytic domain that resembles the classic GT-B motif of nucleotide activated glycosyltransferases. However, the enzyme forms an unprecedented dodecamer in solution and, remarkably, in the asymmetric unit of the crystal structure. The authors show that a beta-hairpin thumb is essential for catalysis and dimerization and subsequent dodecamerization. They also prepared structure-guided mutants and analyzed the effects on catalysis. One of the mutants resulted in a cocrystal structure with ADP-D, D-heptose, which supported the proposed roles of several amino acids. The authors also describe a number of additional interesting features of the structure including a unique iron finger motif that contributes to back to back dimerization within the dodecamer. Finally, they used cryo-EM to analyze the autotransporter TibA and the complex of TibA and TibC, suggesting the complex consists of 12 TibC and 6 TibA molecules. The reviewers all agree that these are exciting findings that provide a significant contribution to the glycosyltransferase/carbohydrate, autotransporter, and post-translational modification fields that warrant publication in *eLife*.

The reviewers were much less enthusiastic about the MD simulations and the strong conclusions drawn from them. The suggested protrusion mechanism for the glycosylation acceptor sites is very speculative and even the notion that the enzyme indeed performs processive catalysis is not supported by any biochemical data. This part of the manuscript is not needed for an otherwise very good story and insufficiently supported at present for publication. It is requested to either remove the model or gather much more experimental data (structural and kinetic) in support of a processive, screw propelling mechanism.

The reviewers request you to address the questions listed below in your manuscript: 1) The authors need to include B-factors for the protein and ligand ADP-D, D-heptose in the data collection table, especially for the low-resolution data set of the ligand complex.

2) The authors need to test whether the TibC mutant that was amenable to crystallization has the identical enzymatic activity of the parent.

3) The authors need to include analytical ultracentrifugation or SEC-multi-angle light scattering in addition to their gel filtration data to determine in a more quantitative manner the molecular weights in solution for TibC alone and the TibA-TibC complex.

4) The authors should, in addition to the 2FoFc map, also include an FoFc omit map of the ligand-binding site with the ligand excluded from the refinement (Figure 6). In Figure 6 the distances of the Pro-ligand and Ser-Ligand should be measured and displayed.

5) The authors mention use of a tandem MS experiment, but they list it as data not shown. The authors need to show the data as it will aid future investigations in other labs.

6) The authors need to describe in more detail how they know the metal is iron (color alone is not definitive). They should provide ICP-MS and/or SAD data. In addition, the refined B-factors of the irons should be listed.

7) The authors show two stereoisomers of nucleotide-heptose are accepted by AAH. It will be important to know if one is used much more efficiently as it could have an impact on the identity of the heptoses on AIDA_I in DAEC.

---

## [Author Response]

We have performed all the suggested experiments and fully addressed the comments made by the reviewers. Meanwhile, we would like to bring to your attention that about a week ago we published a sister article complementary to the current study in *Cell Host & Microbe*. In that paper (*Cell Host & Microbe*, 16, 351–363), we focus on the biochemical and biological functions (in both cell culture and mouse infection) of AAH heptosylation of AIDA-I autotransporter and biochemically define and characterize the BAHT family of heptosyltransferase. The main points of the *Cell Host & Microbe* paper do not overlap with the current manuscript that is more focused on the structure of TibC alone and in complex with the sugar donor as well as the TibA substrate. The two papers illustrate a complete story of the BAHT family of heptosyltransferase from genetic and physiological function to biochemical characterization and structural mechanism. The two papers, coming out around the same time, will generate a bigger impact in the field of bacterial pathogenesis and protein glycosylation.

*The reviewers were much less enthusiastic about the MD simulations and the strong conclusions drawn from them. The suggested protrusion mechanism for the glycosylation acceptor sites is very speculative and even the notion that the enzyme indeed performs processive catalysis is not supported by any biochemical data. This part of the manuscript is not needed for an otherwise very good story and insufficiently supported at present for publication. It is requested to either remove the model or gather much more experimental data (structural and kinetic) in support of a processive, screw propelling mechanism*.

Following the reviewer’s particular suggestion, we have removed the MD simulation part (Figure 8 and its supplement) and also the model for processive heptosylation mechanism (Figure 9). Despite that we have compromised to satisfy the reviewers, we would like to emphasize that our MD simulation was performed at a highest standard. We are very confident that the results are of sufficient quality, and insightful and stimulating for the enzyme catalysis field. We will submit that part of the story separately. However, I do understand that a hard-core experimental biologist may not appreciate the computational results that much.

*1) The authors need to include B-factors for the protein and ligand ADP-D, D-heptose in the data collection table, especially for the low-resolution data set of the ligand complex*.

As requested by the reviewers, we have updated the crystallography table, in which the B-factors for the protein and ligand are included. The refined B-factors for the iron are also listed in response to another comment raised by the reviewer. It is important to note that the B factor for the protein ligand complex is arbitrarily assigned and fixed to 150 for the final refinement because the relatively resolution of the TibC/ADP-heptose complex (3.9 Å), which is a general practice in protein crystallography. The reason we chose this method for the low resolution refinement is because fixed B-factor refinement could reduce the unfavorable ratio of parameters (degrees of freedom) vs data to avoid the potential overfitting problem. In fact, it is a common sense that the atomic B-factors have little significance to reflect the internal thermal motion of atoms when the resolution falls below 3.5 Å. In practice, this method produces the lowest Rfree. Accordingly, we have added a REMARK card to the deposited coordinate file to indicate this.

*2) The authors need to test whether the TibC mutant that was amenable to crystallization has the identical enzymatic activity of the parent*.

Done. In the revised Figure 1 and C, we now demonstrate that TibC mutant used for crystallization has the identical enzymatic activity and biological function as the wild type protein.

*3) The authors need to include analytical ultracentrifugation or SEC-multi-angle light scattering in addition to their gel filtration data to determine in a more quantitative manner the molecular weights in solution for TibC alone and the TibA-TibC complex*.

Done. We have performed the requested analytical ultracentrifugation experiments. The data included as new Figure 2—figure supplement 2 show the measured molecular masses of both TibC alone and TibA-TibC complex, which are consistent with the gel filtration data.

*4) The authors should, in addition to the 2FoFc map, also include an FoFc omit map of the ligand-binding site with the ligand excluded from the refinement (*Figure 6*). In*
Figure 6
*the distances of the Pro-ligand and Ser-Ligand should be measured and displayed*.

Done. Please see the revised Figure 6, which includes an FoFc omit map of the ligand-binding site (Figure 6) and the distances of the Pro-ligand and Ser-Ligand (Figure 6).

*5) The authors mention use of a tandem MS experiment, but they list it as data not shown. The authors need to show the data as it will aid future investigations in other labs*.

Done. We have included the tandem MS data as the new Figure 1.

*6) The authors need to describe in more detail how they know the metal is iron (color alone is not definitive). They should provide ICP-MS and/or SAD data. In addition, the refined B-factors of the irons should be listed*.

We have recently published a sister paper in *Cell Host & Microbe*, in which we demonstrate the biochemical activity of AAH and more importantly, the physiological function of AIDA-I heptosylation in a mouse infection model. In that paper, we did perform ICP-MS analysis and demonstrate that the color results from the iron in TibC protein. We have cited the *Cell Host & Microbe* paper to indicate this information. Moreover, our TibC crystal structure was solved by using anomalous signals from the bound ferric ion, which confirms from another angel that the metal is iron. The B-factor for iron has also been included in the revised Table 1.

*7) The authors show two stereoisomers of nucleotide-heptose are accepted by AAH. It will be important to know if one is used much more efficiently as it could have an impact on the identity of the heptoses on AIDA_I in DAEC*.

In response to the reviewers’ comments, we have titrated the amounts of AAH for both ADP-D, D-heptose and the anomer ADP-L, D-heptose ligands in the *in vitro* enzymatic assay. The data included as Figure 6—figure supplement 2 show that AAH has a slight preference for ADP-L, D-heptose vs ADP-D, D-heptose. As a continuous enzymatic assay allowing for quantitative measurements of enzyme kinetics has not been established for this kind of novel enzymatic activity, the Western blot-based titration data is only of an indication nature, but does provide some useful information for future studies.